# One-Transistor Dynamic Random-Access Memory Based on Gate-All-Around Junction-Less Field-Effect Transistor with a Si/SiGe Heterostructure

**Young Jun Yoon** [1] , **Jae Sang Lee** [1] , **Dong-Seok Kim** [1] , **Sang Ho Lee** [2] **and In Man Kang** [2,*]

[1]  Korea Multi-purpose Accelerator Complex, Korea Atomic Energy Research Institute, Gyeongju 38180, Korea; yjyoon@kaeri.re.kr (Y.J.Y.); jslee8@kaeri.re.kr (J.S.L.); dongseokkim@kaeri.re.kr (D.-S.K.)
[2]  School of Electronic and Electrical Engineering, Kyungpook National University, Daegu 41566, Korea; jim782jim@naver.com
*  Correspondence: imkang@ee.knu.ac.kr; Tel.: +82-053-950-5513

**Abstract:** This paper presents a one-transistor dynamic random-access memory (1T-DRAM) cell based on a gate-all-around junction-less field-effect transistor (GAA-JLFET) with a Si/SiGe heterostructure for high-density memory applications. The proposed 1T-DRAM achieves the sensing margin using the difference in hole density in the body region between '1' and '0' states. The Si/SiGe heterostructure forms a quantum well in the body and reduces the band-to-band tunneling (BTBT) barrier between the body and drain. Compared with the performances of the 1T-DRAM with Si homo-structure, the proposed 1T-DRAM improves the sensing margin and retention time because its storage ability is enhanced by the quantum well. In addition, the thin BTBT barrier reduced the bias condition for the program operation. The proposed 1T-DRAM showed a high potential for memory applications by obtaining a high read current ratio at '1' and '0' states about $10^8$ and a long retention time above 10 ms.

**Keywords:** junction-less field-effect transistor; gate-all-around; one-transistor dynamic random-access memory; Si/SiGe heterostructure

## 1. Introduction

As the electronics industry continuously expands through various applications, such as smart devices, cloud computing, artificial intelligence, and the Internet of things (IoT), the demand for the scaling of dynamic random-access memory (DRAM) for high-density storage and high-performance rises [1–5].

However, DRAMs typically consist of one-transistor (1T) and one-capacitor (1C). Such devices have a scaling limitation because the relatively large size of the capacitor cannot be reduced while maintaining its charge retention characteristics. Therefore, one-transistor dynamic random-access memory (1T-DRAMs), sometimes called capacitorless DRAMs, because the memory operation is realized using the floating body effect instead of the capacitor, have been studied for next-generation DRAM technology [6–10]. Both a silicon-on-insulator (SOI) single-gate structure [11,12] and a double-gate structure [13,14] can be employed for realizing a 1T-DRAM. However, these structures have many limitations in terms of integration density, price competitiveness, and compatibility with the fabrication process.

In this paper, we present a 1T-DRAM based on a gate-all-around junction-less field-effect transistor (GAA-JLFET) with a Si/SiGe heterostructure. The GAA-JLFET is one of the main next-generation structures for improving the chip integration density and performance. The Si/SiGe heterojunction was employed to improve the charge storage ability and reduce the bias conditions of the program operation.

The memory characteristics such as the sensing margin and retention time were obtained using the 3-dimensional (3D) technology computer-aided design (TCAD) simulator and were investigated to verify their potential for memory applications.

## 2. Device Structure and Simulation Method

Figure 1a shows a 3D schematic and cross-sectional diagram of the GAA-JLFET-based 1T-DRAM. The round nanowire had a radius of 10 nm and consisted of the n-type Si/SiGe heterostructure. A low Ge content in SiGe of 0.2 was determined from the lattice mismatch between Si and SiGe. Doping concentrations of Si drain, SiGe body, and Si source regions were $5 \times 10^{19}$ cm$^{-3}$, $1 \times 10^{18}$ cm$^{-3}$, and $5 \times 10^{19}$ cm$^{-3}$, respectively. The body region was depleted by a high-gate work-function of 5.0 eV at a gate-to-source voltage ($V_{GS}$) of 0 V (off-state) [15,16]. The depleted body region was used as a hole charge storage area. A valence band offset formed across the Si/SiGe interface. In general, the band alignment of the Si/SiGe heterojunction make a high valence band offset [17]. As shown in Figure 1b, the 1T-DRAM obtained a deep quantum well in the SiGe body region induced by a high valence band offset. It is important because the quantum well can improve the retention characteristics of a 1T-DRAM [18]. The thickness of the gate dielectric HfO$_2$ was 2 nm. We designed a long channel length of 100 nm. A long channel length can improve a storage ability because a size of storage area is related to a channel length of the proposed 1T-DRAM. Therefore, the integration density of the 1T-DRAM can be improved by designing vertical GAA or stacked lateral GAA structures. The gate length ($L_G$) and underlap length ($L_{underlap}$) were 70 nm and 15 nm, respectively. The role of the underlap structure was to reduce the electric field generated at the body/drain and source/body interfaces, which arose due to the Shockley-Read-Hall (SRH) and trap-assisted tunneling (TAT) recombination [19–21]. The underlap structure can increase the biases for band-to-band tunneling (BTBT), which was used for forming electron-hole pairs (EHPs) as the program principle in the 1T-DRAM. The Si/SiGe heterojunction reduced the bias conditions for the program operation because its high valence band offset can induce a higher BTBT rate than a Si homojunction [22,23].

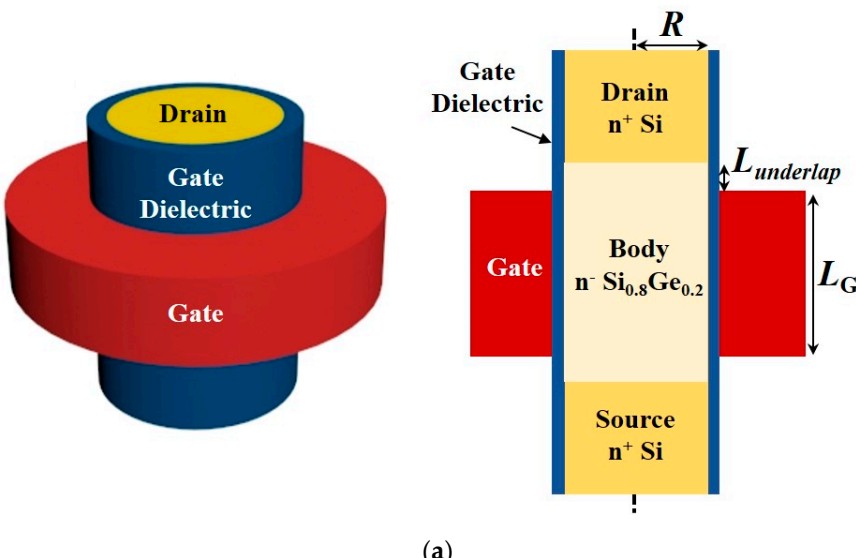

(a)

**Figure 1.** *Cont.*

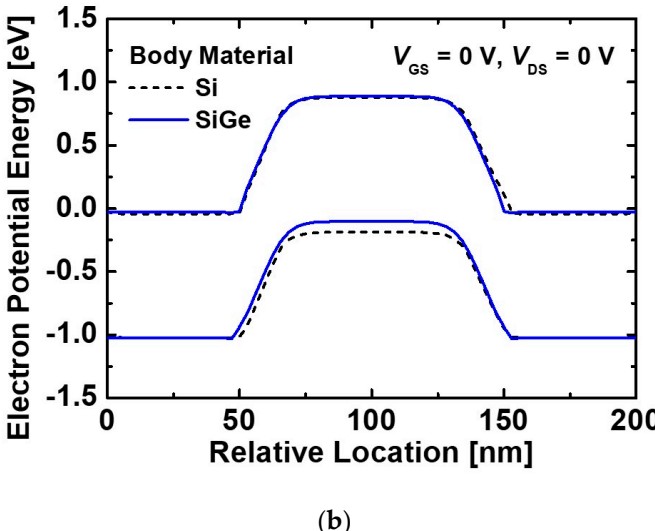

(**b**)

**Figure 1.** (**a**) 3D schematic and cross-section of the GAA-JLFET-based 1T-DRAM. (**b**) Energy band diagrams under the gate dielectric of the GAA-JLFET-based 1T-DRAMs with different body materials at $V_{GS}$ of 0 V and $V_{DS}$ of 0 V.

In order to evaluate the memory performance, a simulation was carried out using TCAD Sentaurus, which is the simulation package offered by Synopsys to develop and optimized semiconductor devices [24]. Sentaurus TCAD is capable of simulating numerically electrical and thermal characteristics of devices. Various physical models, such as SRH recombination, TAT, BTBT, and quantum confinement effects, were employed in the simulation to obtain reliable characteristics of 1T-DRAMs. In SRH and TAT models, the electron and hole lifetimes in Si were $1 \times 10^{-5}$ s and $3 \times 10^{-6}$ s, respectively [25]. In case of SiGe, the electron lifetime of $5 \times 10^{-6}$ s and hole lifetime of $5 \times 10^{-7}$ s were set because the smaller recombination lifetime in SiGe than that in Si can be caused the by growth processes [26]. The lifetimes in Si and SiGe were significantly reduced by the doping concentration and inner electric field.

## 3. Results and Discussion

Figure 2 shows the transfer curves of the 1T-DRAM at the temperatures of 300 K and 358 K. The 1T-DRAM exhibited a low off-state current ($I_{off}$) of about $10^{-16}$ A/µm and a high threshold voltage ($V_{th}$) at a drain-to-source voltage ($V_{DS}$) of 0.1 V. These results indicate that the body region was fully depleted by the gate metal at a $V_{GS}$ of 0 V. Here, full depletion can improve the retention characteristics of the 1T-DRAM because it reduces the recombination rate. When the temperature rose from 300 K to 358 K, the $I_{off}$ increased because a high temperature leads to faster carrier generation. An increase in carrier density can degenerate the retention characteristics due to the increase in recombination.

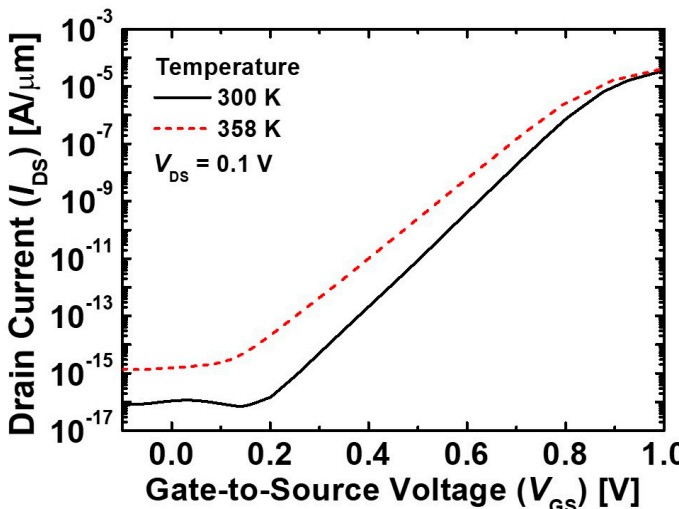

**Figure 2.** Transfer curves of the GAA-JLFET-based 1T-DRAM at temperatures of 300 K and 358 K.

Figure 3 shows the transient characteristics of the 1T-DRAMs with Si and SiGe bodies for memory operations at the temperature of 300 K. The bias conditions for the operation states are summarized in Table 1. First, the program (write '1') operation was performed using BTBT between the body and drain regions to set the 1T-DRAMs to '1' state, as shown in Figure 4a. The thin tunneling barrier was formed when voltages $V_{G\_P} = -1.0$ V and $V_{D\_P} = 1.5$ V were applied at the gate and drain, respectively, as shown in Figure 4b. The tunneling barrier width was reduced by the Si/SiGe heterojunction. The holes generated by BTBT were stored under the valence band of the body region, while the generated electrons flowed through the drain region. The '1' state of 1T-DRAM was completed by storing the hole charges in the body region. The read current at '1' state ($I_{R1}$) of the 1T-DRAM with the Si/SiGe heterojunction increased, whereas $I_{R1}$ of the DRAM with the Si homojunction was almost unchanged. This is due to the difference of the tunneling barrier width. The 1T-DRAM with the Si homojunction obtained a small hole density after writing a '1' operation because a low BTBT was induced by a relatively thick tunneling barrier. Thus, the 1T-DRAM with the Si homojunction needs a higher bias to write the '1' operation.

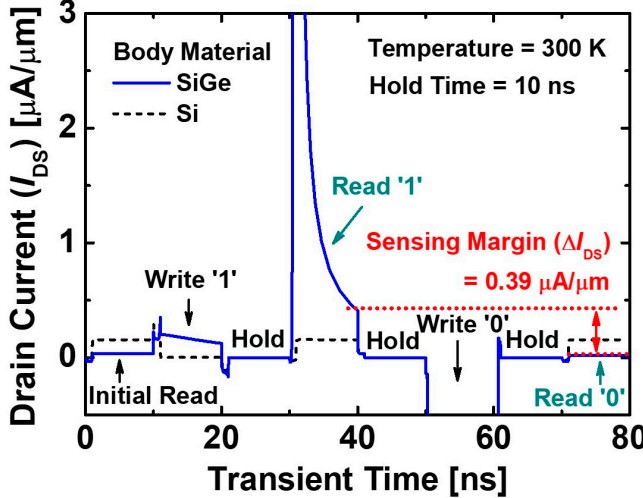

**Figure 3.** Transient characteristics of the GAA-JLFET-based 1T-DRAMs with different body materials at the temperature of 300 K. The read/hold/write operation time is 10 ns.

**Table 1.** Bias conditions for memory operations.

| Operation | Write '1' | Write '0' | Read | Hold |
|---|---|---|---|---|
| Gate Voltage ($V_G$) [V] | −1.0 | 1.0 | 0.7 | 0 |
| Drain Voltage ($V_D$) [V] | 1.5 | −1.5 | 0.1 | 0 |

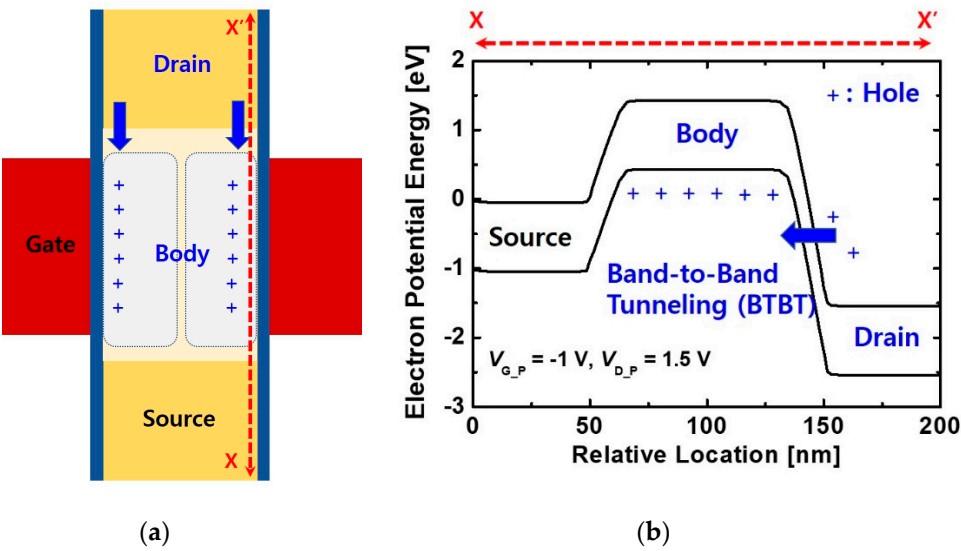

(**a**)

(**b**)

**Figure 4.** (**a**) A schematic diagram and (**b**) the energy band diagram for the write '1' operation. $V_{G\_P}$ and $V_{D\_P}$ are −1.0 V and 1.5 V, respectively.

It can be removed with the erase (write '0') operation, as shown in Figure 5a. Here, the energy band of the drain was raised by the applied gate voltage ($V_{G\_E}$) of 1.0 V and the drain voltage ($V_{D\_E}$) of −1.5 V, as shown in Figure 5b. The holes stored in the body region moved through the drain region because the potential barrier for the hole was removed by the influence of the negative $V_{D\_E}$ value. After the write '0' operation, the hole density in the body region of the 1T-DRAM was smaller than that at the '1' state.

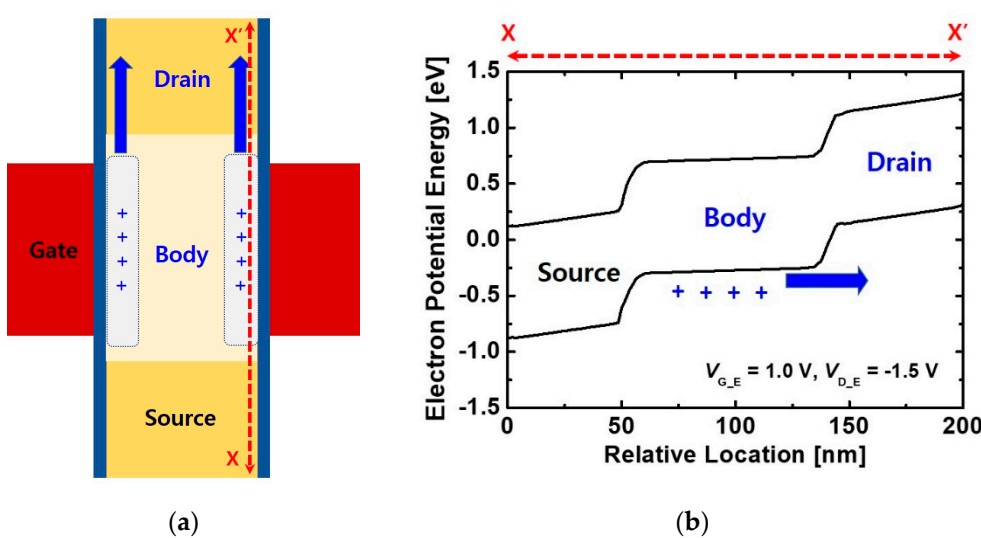

(**a**)

(**b**)

**Figure 5.** (**a**) A schematic diagram and (**b**) the energy band diagram for the write '0' operation. $V_{G\_E}$ and $V_{D\_E}$ are 1.0 V and −1.5 V, respectively.

Here, the difference in hole density between '1' and '0' states induced the changing depletion region in the center of the body, as shown in Figure 6a. At the '1' state, the potential barrier height was reduced because the stored holes removed the depletion area from the center of the body, as shown in Figure 6b. Variations in the current were caused by the difference in the potential barrier heights at '1' and '0' states. As shown by the electron and hole density distribution in Figure 6b,c, the electron density in the center of the body at the '1' state is higher than that at the '0' state because the depletion region was reduced by a high hole density. Both hole and electron densities at the '0' state were low. The $I_{R1}$ of 1T-DRAM is higher than that at the '0' state ($I_{R0}$) due to the lowering of the potential barrier. As a result, the sensing margin of the 1T-DRAM with the SiGe body is 0.39 µA/µm at the bias condition of the read operation (gate voltage $V_{G\_R}$ = 0.7 V and drain voltage $V_{D\_R}$ = 0.1 V). The sensing margin was defined as the difference between $I_{R1}$ and $I_{R0}$. The 1T-DRAM with the SiGe body achieved a higher sensing margin than the 1T-DRAM with the Si body because of the efficiency of BTBT and the presence of the quantum well. In addition, $I_{R1}$ and $I_{R0}$ in the proposed 1T-DRAM are easily distinguishable due to a high $I_{R1}/I_{R0}$ ratio of 22.3.

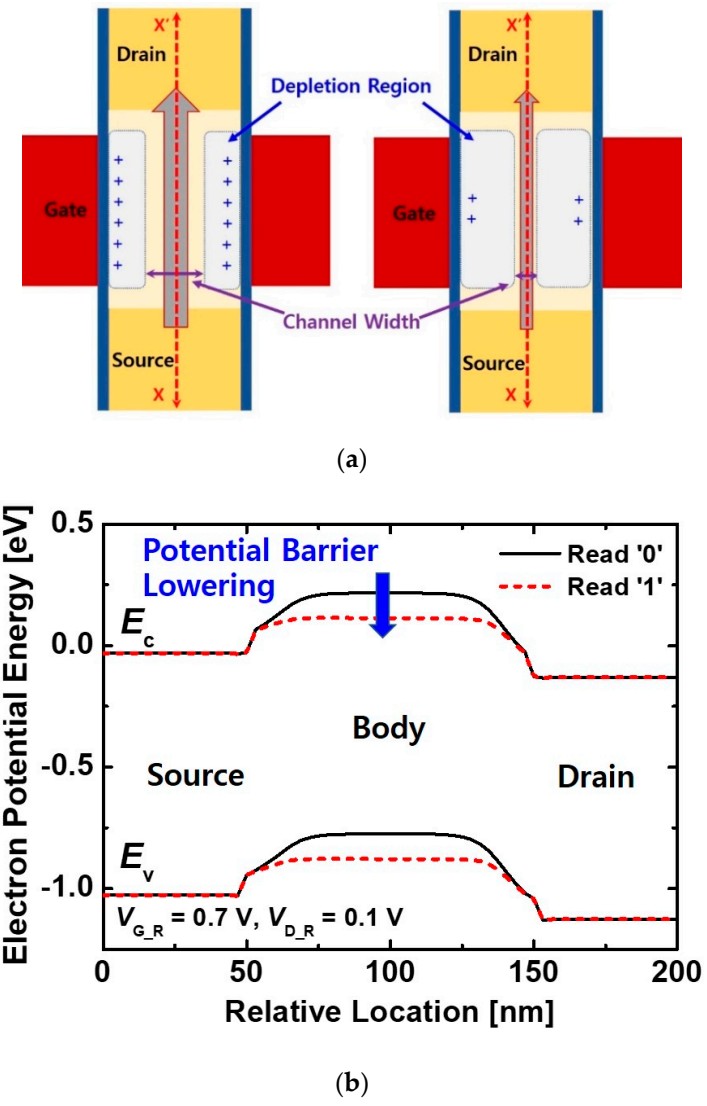

(a)

(b)

**Figure 6.** *Cont.*

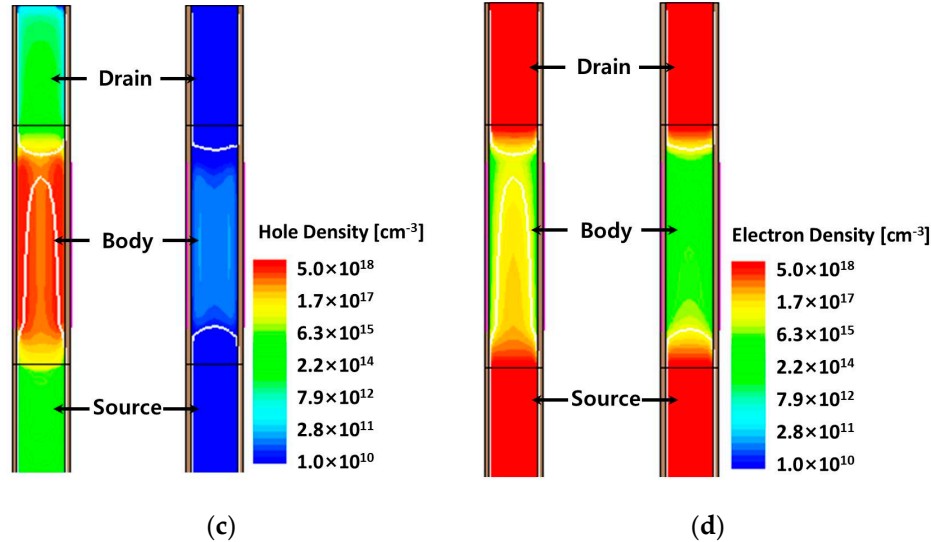

(c)                                          (d)

**Figure 6.** (**a**) A schematic diagram and (**b**) the energy band diagrams for the read '1' and '0' operations. Simulated results for (**c**) the hole density distribution and (**d**) the electron density distribution for the read '1' and '0' operations are also provided. $V_{G\_R}$ and $V_{D\_R}$ are 0.7 V and 0.1 V, respectively.

Figure 7 shows the read currents $I_R$ at '1' and '0' states as functions of $V_{G\_R}$. When $V_{G\_R}$ was 0.2 V, the 1T-DRAM obtained a very high $I_{R1}/I_{R0}$ ratio of about $10^8$. As $V_{G\_R}$ increased, $I_{R0}$ rose gradually because a high $V_{G\_R}$ formed a channel. When a $V_{G\_R}$ above 0.8 V was applied, the difference between $I_{R1}$ and $I_{R0}$ disappeared because the storage region was removed with the depletion region.

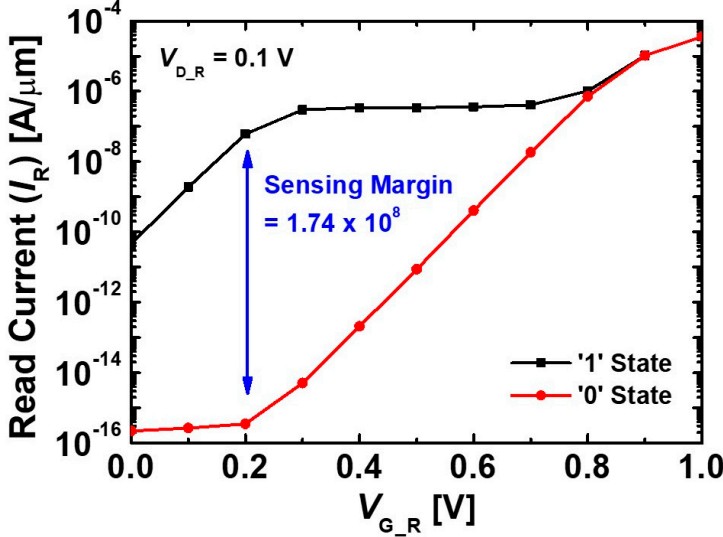

**Figure 7.** Read currents $I_{R1}$ and $I_{R0}$ as functions of $V_{G\_R}$ at the temperature of 300 K.

Figure 8a shows the $I_{R1}$ and $I_{R0}$ of the 1T-DRAM as functions of hold time at the temperatures of 300 K and 358 K. At the former temperature, the $I_{R1}$ and $I_{R0}$ values were maintained for a long time. Two  factors helped to achieve such excellent retention characteristics. The Si/SiGe heterostructure quantum well suppressed the diffusion of the stored hole charges, as shown in Figure 8b, and the underlap structure minimized the carrier recombination and generation across the source/gate and drain/gate interfaces. As hold time increased, $I_{R1}$ decreased because recombination reduced the hole density. The increase in $I_{R0}$ was influenced by carrier generation between the body/drain and body/source interfaces, which was caused by the SRH and TAT mechanisms while holding the '0' state, as shown in Figure 8c. At the temperature of 358 K (approximately 85 °C), there were no significant

changes in the $I_{R1}$ and $I_{R0}$ values up to 10 ms due to the reduction in carrier recombination and generation rates caused by the underlap structure.

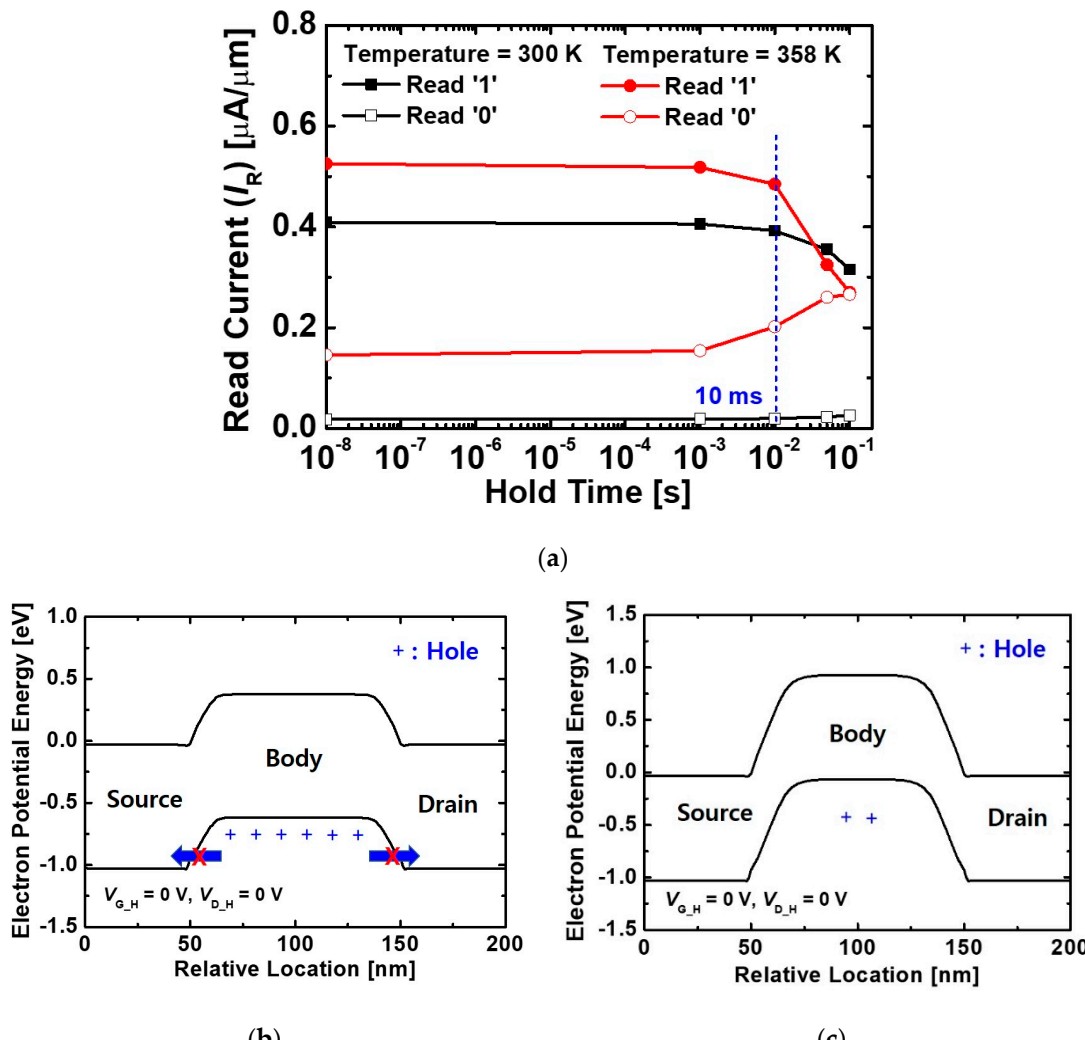

**Figure 8.** (**a**) Read currents $I_{R1}$ and $I_{R0}$ as functions of hold time at the temperatures of 300 K and 358 K. (**b**) The energy band diagrams for the hold operation at (**b**) '1' and (**c**) '0' states are also provided. The $V_{G\_H}$ and $V_{D\_H}$ are both at 0 V.

The retention characteristics of the proposed 1T-DRAM are better than those of the 1T-DRAM with a Si body, as shown in Figure 9a. Here, the retention time is defined as the time elapsed until the sensing margin reaches 50% of its initial value. When the temperature was 300 K, the 1T-DRAM with a SiGe body retained the sensing margin for a hold time above 100 ms, whereas the sensing margin of the 1T-DRAM with a Si body sharply decreased after a hold time of 1 ms. The retention characteristics of the 1T-DRAM was affected by the diffusion and recombination mechanisms of the stored holes. The recombination rate in the proposed 1T-DRAM was higher than that of the 1T-DRAM with a Si body due to the high stored hole density, as shown in Figure 9b. However, the proposed 1T-DRAM obtained a long retention time because the diffusion of the holes was suppressed by the valence band offset between Si and SiGe. As a result, the diffusion mechanisms were the dominating factors reducing the retention time. Although the sensing margin was reduced at the temperature of 358 K, the proposed 1T-DRAM achieved a retention time above 10 ms. These results demonstrate that the proposed device has a high potential for embedded memory applications.

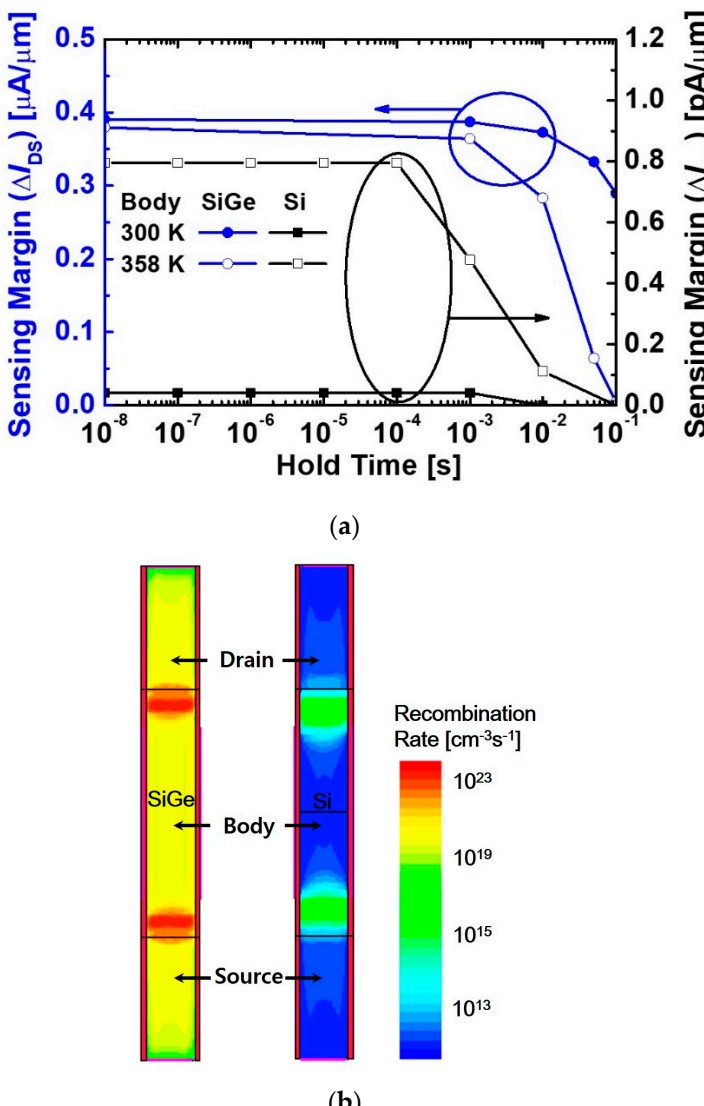

**Figure 9.** (**a**) Comparison of sensing margins as functions of hold time between 1T-DRAMs with Si and SiGe body at temperatures of 300 K and 358 K. (**b**) Recombination rates in 1T-DRAMs with Si and SiGe body for the hold operation at the '1' state (hold time = 10 ns) and temperatures of 300 K.

The performances of the 1T-DRAM with the Si/SiGe heterojunction were substantially affected by the valence band offset between Si and SiGe, which can be changed by the variation of Ge content in SiGe. In addition, we investigated the effects of Ge content in the SiGe body on the sensing margin and retention time of the 1T-DRAM, as shown in Figure 10. With the increase in Ge content, the sensing margin of the 1T-DRAM drastically increased. This result was due to a high BTBT rate, which was formed by the increased valence band offset between Si and SiGe. When the higher BTBT occurs, the sensing margin was improved by a higher hole density. In terms of retention time, the 1T-DRAM with high Ge content of 0.4 exhibited a short retention time of 7.7 ms. Although the high valence band offset between Si and SiGe suppressed the hole diffusion, the recombination rate became high at the source/body and body/drain interfaces. As a result, the increase in the recombination rate degraded the retention time of the 1T-DRAM with high Ge content. Thus, an optimum Ge content in the SiGe body was 0.3, considering the memory characteristics and lattice mismatch between Si and SiGe. The variation of Ge content in the proposed 1T-DRAM induced the trade-off between the sensing margin and retention time. The 1T-DRAMs obtained the trade-off between the sensing margin and retention time due to design parameters, such as channel doping [27] and body thickness [28],

which affected the stored hole density and recombination rate simultaneously. Therefore, the memory characteristics of the proposed 1T-DRAMs can be improved by optimizing the design parameters.

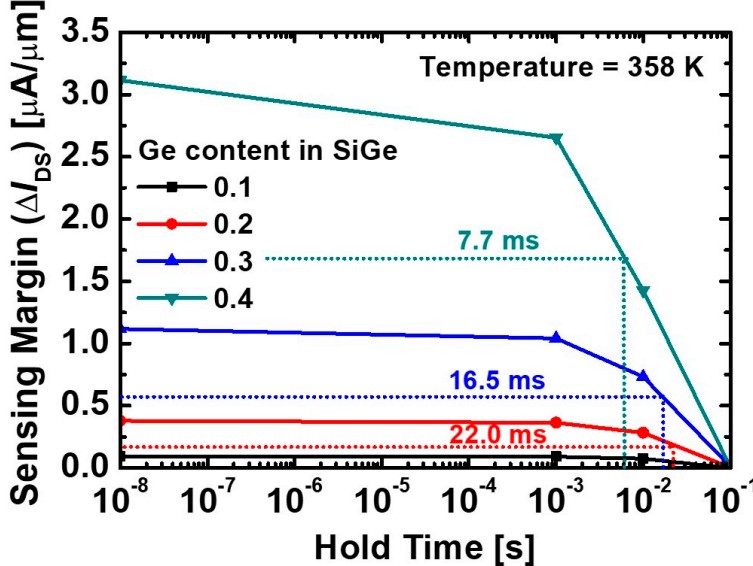

**Figure 10.** Effects of Ge content in the SiGe body on sensing margins and retention times of the 1T-DRAMs at a temperature of 358 K.

## 4. Conclusions

The GAA-JLFET-based 1T-DRAM cell with the Si/SiGe heterostructure was presented for high-density memory applications. The efficiency of program operations was realized by BTBT between the SiGe body and Si drain regions due to the reduction in the tunneling barrier. The proposed 1T-DRAM achieved a high ratio of $I_{R1}/I_{R0}$ of about $10^8$ at a low $V_{G\_R}$ of 0.2 V and $V_{D\_R}$ of 0.1 V. The retention time above 10 ms was obtained at the temperature of 358 K due to the quantum well, which was formed by the valence band offset between Si and SiGe. As a result, the proposed 1T-DRAMs is a suitable memory cell for high-density applications because of its nanowire structure and excellent memory performance.

**Author Contributions:** Conceptualization, Y.J.Y. Investigation, Y.J.Y. Data analysis, Y.J.Y., J.S.L., D.-S.K., S.H.L. and I.M.K. Writing—original draft preparation, Y.J.Y. Writing—review and editing, I.M.K. All authors have read and agreed to the published version of the manuscript.

**Funding:** This work was supported by the National Research Foundation of Korea (NRF) grant funded by the Korea government (MSIT) (No. NRF-2020R1A2C1005087) and in part by Samsung Electronics Co. Ltd. The study was supported by the 4 BK21 project funded by the Ministry of Education, Korea, by the Ministry of Trade, Industry & Energy (MOTIE) (10080513), and Korea Semiconductor Research Consortium (KSRC) support program for developing the future semiconductor devices. This work was also supported by the KOMAC (Korea Multi-purpose Accelerator Complex) operation fund of KAERI (Korea Atomic Energy Research Institute).

**Conflicts of Interest:** The authors declare no conflict of interest.

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
