# Peer review of "One-Transistor Dynamic Random-Access Memory Based on Gate-All-Around Junction-Less Field-Effect Transistor with a Si/SiGe Heterostructure"

_electronics, doi:10.3390/electronics9122134_

Round 1

Reviewer 1 Report

The way to modify parameters of the gate-all-around junctionless field-effect transistor (GAA JL-NWFET) is suggested. The channel of the FET was suggested to form as а heterostructure by doping the silicon with germanium. The quantum well thus formed enhances the retention properties of the memory cell in comparison with all-Si FET structure by one order of the retention time.

It can be interesting, however there are some flaws, which should be considered before publication.

1. The TCAD software used is unspecified. The results shown may be prospective and useful, however they should be backed by the detailed specification of the TCAD simulator used.
2. The design suggested is considered with an only parameters combination, except of a temperature, while in can be simulated with different doping levels, different geometry parameters, etc. I believe that possible parameters should be considered and compared in larger number to prove that the presented variant is the best.

Author Response

Comments to the Author
The way to modify parameters of the gate-all-around junctionless field-effect transistor (GAA JL-NWFET) is
suggested. The channel of the FET was suggested to form as а heterostructure by doping the silicon with
germanium. The quantum well thus formed enhances the retention properties of the memory cell in comparison
with all-Si FET structure by one order of the retention time.
It can be interesting, however there are some flaws, which should be considered before publication.

1) The TCAD software used is unspecified. The results shown may be prospective and useful, however they should
be backed by the detailed specification of the TCAD simulator used.

Answer)
Thank you for your comments. We specified the used TCAD software in the manuscript. And, we added the detailed specification of the TCAD simulator.
-> “In order to evaluate the memory performance, a simulation was carried out using TCAD Sentaurus, which is the simulation package offered by Synopsys to develop and optimized semiconductor devices [24]. Sentaurus TCAD is capable is simulating numerically electrical and thermal characteristics of devices. Various physical models, such as SRH recombination, TAT, BTBT, and quantum confinement effects, were employed in the simulation to obtain reliable characteristics of 1T-DRAMs. In SRH and TAT models, the electron and hole lifetimes in Si were 1×10-5
s and 3×10-6 s, respectively [25]. In case of SiGe, the electron lifetime of 5×10-6 s and hole lifetime of 5×10-7 s were set because the smaller recombination lifetime in SiGe than that in Si can be caused the by growth processes [26]. The lifetimes in Si and SiGe were significantly reduced by the doping concentration and inner electric field.”
(68th line – 80th line, page 3)

2) The design suggested is considered with an only parameters combination, except of a temperature, while in can be simulated with different doping levels, different geometry parameters, etc. I believe that possible parameters should be considered and compared in larger number to prove that the presented variant is the best.

Answer)
We expected that the Ge content in SiGe was one of important factors which significantly affected memory characteristics because the energy band of the body region can be changed the variations of Ge content in SiGe. Thus, we added the data and sentences for sensing margin and retention time of 1T-DRAMs with different Ge content.

-> “The performances of the 1T-DRAM with Si/SiGe heterojunction were substantially affected by the valence band offset between Si and SiGe, which can be changed by the variation of Ge content in SiGe. In addition, we investigated the effects of Ge content in SiGe body on the sensing margin and retention time of the 1T-DRAM, as shown in Fig. 10. With the increase in Ge content, the sensing margin of the 1T-DRAM drastically increased. This result was due to a high BTBT rate, which was formed by the increased valence band offset between Si and SiGe. When the higher BTBT occurs, the sensing margin was improved by a more high hole density. In terms of retention time, the 1T-DRAM with high Ge content of 0.4 exhibited a short retention time of 7.7 ms. Although the high valence band offset between Si and SiGe suppressed the hole diffusion, the recombination rate became high the recombination rate became high at the source/body and body/drain interfaces. As a result, the increase in recombination rate degraded the retention time of the 1T-DRAM with high Ge content. Thus, an optimum Ge content in SiGe body was 0.3 considering the memory characteristics and lattice mismatch between Si and SiGe.”
(181th line – 192th line, page 9)

Reviewer 2 Report

A GAA-JLFET-based 1T-DRAM cell with the Si/SiGe heterostructure was studied with 3D TCAD simulation. Although most results are reasonable and may be useful, some revision and addition are advised.

  1. The energy band diagrams in Fig. 1(b) may be wrong. The valence band edge (Ev) of SiGe should be higher than that of Si. Also, the potential barrier at source/body and body/drain are likely too large ( ~ 1 eV).
  2. A Ge content in SiGe of 0.2 was used. Since a higher Ge content like 0.3 in SiGe is also generally used, the memory performance of 1T-DRAM cell with Ge f 0.3 in the Si/SiGe heterostructure may be better. Please comment and add the data.
  3. The channel length of device in this work is ~ 100 nm, which is quite long as compared to the-state-of-the-art. Could a device with channel length < 30 nm be studied ? How is the difference ?
  4. Several English grammar errors should be corrected throughout the manuscript, such as,

- The proposed 1T-DRAM achieved the sensing margin using the difference in hole density in the body region between ‘1’ and ‘0’ states. -The Si/SiGe heterostructure formed a quantum well in the body and reduced the band-to-band tunneling (BTBT) barrier between the body and drain… achieved, formed, reduced => achieves, forms, reduces

-Compared with the performances of the 1T DRAM with Si homostructure, the proposed 1T-DRAM were improves on the …  were => typo

Author Response

Comments to the Author
A GAA-JLFET-based 1T-DRAM cell with the Si/SiGe heterostructure was studied with 3D TCAD simulation. Although most results are reasonable and may be useful, some revision and addition are advised.

1) The energy band diagrams in Fig. 1(b) may be wrong. The valence band edge (Ev) of SiGe should be higher
than that of Si. Also, the potential barrier at source/body and body/drain are likely too large ( ~ 1 eV).

Answer)
Thank you for your comments. In Fig. 1(b), we made a mistake for the legend and revised the legend. In case of the potential barrier, the JLFET exhibited a high potential barrier height in energy band under gate dielectric when VGS is 0 V because the JLFET consisted of the gate with a high work-function, which fully depleted the body region. Thus, the potential barrier was large.

2) A Ge content in SiGe of 0.2 was used. Since a higher Ge content like 0.3 in SiGe is also generally used, the memory performance of 1T-DRAM cell with Ge of 0.3 in the Si/SiGe heterostructure may be better. Please comment and add the data.

Answer)
In order to investigate the effects Ge content in SiGe on performance, we added the figure and sentences for the
sensing margins and retention times of 1T-DRAMs with different Ge content.

-> “The performances of the 1T-DRAM with Si/SiGe heterojunction were substantially affected by the valence band offset between Si and SiGe, which can be changed by the variation of Ge content in SiGe. In addition, we investigated the effects of Ge content in SiGe body on the sensing margin and retention time of the 1T-DRAM, as shown in Fig. 10. With the increase in Ge content, the sensing margin of the 1T-DRAM drastically increased. This result was due to a high BTBT rate, which was formed by the increased valence band offset between Si and SiGe. When the higher BTBT occurs, the sensing margin was improved by a more high hole density. In terms of retention time, the 1T-DRAM with high Ge content of 0.4 exhibited a short retention time of 7.7 ms. Although the high valence band offset between Si and SiGe suppressed the hole diffusion, the recombination rate became high at the source/body and body/drain interfaces. As a result, the increase in recombination rate degraded the retention time of the 1T-DRAM with high Ge content. Thus, an optimum Ge content in SiGe body was 0.3 considering the memory characteristics and lattice mismatch between Si and SiGe.”
(181th line, page 8 – 192th line, page 9) 

3) The channel length of device in this work is ~ 100 nm, which is quite long as compared to the-state-of-the-art. Could a device with channel length < 30 nm be studied ? How is the difference ?

Answer)
We agreed the reviewer opinion. The device had a long channel length of 100 nm compared to the state-of-the art. Because the channel length of proposed device affected storage ability, the device with short channel length of 30 nm obtains the shorter retention time than that of the device with a long channel. However, since the device can be applied in vertical structure or stacked GAA structure, we think that there are no problem in terms of integration density. The sentence for channel length and integration were added in the manuscript.
-> “We designed a long channel length of 100 nm. A long channel length can improve a storage ability because a size of storage area is related to a channel length of the proposed 1T-DRAM. Therefore, the integration density of the 1T-DRAM can be improved by designing vertical GAA or stacked lateral GAA structures.”
(60th line, page 2 – 64th line, page 3) 

4) Several English grammar errors should be corrected throughout the manuscript, such as,

- The proposed 1T-DRAM achieved the sensing margin using the difference in hole density in the body region
between ‘1’ and ‘0’ states. -The Si/SiGe heterostructure formed a quantum well in the body and reduced the band￾to-band tunneling (BTBT) barrier between the body and drain… achieved, formed, reduced => achieves, forms, reduces

-Compared with the performances of the 1T DRAM with Si homostructure, the proposed 1T-DRAM were
improves on the … were => typo

Answer)
Thank you for kind comments. We revised English grammar errors. In addition, we are going to use English grammar revision service supplied from MDPI journal.

Reviewer 3 Report

This paper discussed the performance of a 1T-DRAM cell based on a GAA-JLFET with a Si/SiGe heterostructure.  3D TCAD simulator was used to verify the memory applications.  Several concerns are given for the authors to address before publication.

  1. In Fig.1 (b), the energy band diagrams of Si and SiGe were shown for comparison. The bandgap of SiGe seems to be larger than Si in this figure.  Please make sure of it.
  2. In Fig.3, the transient behaviors of Si and SiGe were compared. The transient Read 1 current of SiGe is significant but not of Si.  Please describe the main reason of this difference.
  3. The net recombination rates between Si and SiGe are of interest for their comparison of transient behaviors. The simulation parameters between Si and SiGe should be desvribed in more detail.
  4. The minority hole carriers are of importance for the transient characteristics. Please give description on the lifetime of hole in this work.  What is the main concern of the control parameter of hole lifetime in this work.

Author Response

Comments to the Author
This paper discussed the performance of a 1T-DRAM cell based on a GAA-JLFET with a Si/SiGe heterostructure. 3D TCAD simulator was used to verify the memory applications. Several concerns are given for the authors to address before publication.

1) In Fig.1 (b), the energy band diagrams of Si and SiGe were shown for comparison. The bandgap of SiGe seems to be larger than Si in this figure. Please make sure of it.

Answer) Thank you for your comments. In Fig. 1(b), we made a mistake for the legend and revised the legend.

2) In Fig.3, the transient behaviors of Si and SiGe were compared. The transient Read 1 current of SiGe is significant but not of Si. Please describe the main reason of this difference.

Answer) We described the main reason of the difference.
-> “The read current at ‘1’ state (IR1) of the 1T-DRAM with the Si/SiGe heterojunction increased, whereas IR1 of the DRAM with the Si homojunction was almost unchanged. This is due to the difference of the tunneling barrier width. The 1T-DRAM with the Si homojunction obtained a small hole density after write ‘1’ operation because a low BTBT was induced by a relatively thick tunneling barrier. Thus, the 1T-DRAM with the Si homojunction needs more high bias for write ‘1’ operation.”
(103th line – 108th line, page 4)

3) The net recombination rates between Si and SiGe are of interest for their comparison of transient behaviors. The simulation parameters between Si and SiGe should be desvribed in more detail. The minority hole carriers are of importance for the transient characteristics. Please give description on the lifetime of hole in this work. What is the main concern of the control parameter of hole lifetime in this work.

Answer) We added the figure and sentences for the mechanisms reducing the stored holes.

-> “In SRH and TAT models, the electron and hole lifetimes in Si were 1×10-5 s and 3×10-6 s, respectively [25]. In case of SiGe, the electron lifetime of 5×10-6 s and hole lifetime of 5×10-7 s were set because the smaller recombination lifetime in SiGe than that in Si can be caused the by growth processes [26]. The lifetimes in Si and SiGe were significantly reduced by the doping concentration and inner electric field.” (76th line – 80th line, page 3)

[25] Schroder, D. K. Carrier Lifetimes in Silicon. IEEE Trans. Electron Devices. 1997, 44, pp. 160-170.
[26] Chang, S. T.; Liu, C. W.; Lu, S. C. Base transit time of graded-base Si/SiGe HBTs considering recombination lifetime and velocity saturation. Solid-State Electron. 2004, 48, pp. 207-215.
(page 11)

“The retention characteristics of the 1T-DRAM was affected by the diffusion and recombination mechanisms of the stored holes. The recombination rate in the proposed 1T-DRAM was higher than that of the 1T-DRAM with a Si body due to the high stored hole density, as shown in Fig. 9(b). However, the proposed 1T-DRAM obtained a long retention time because the diffusion of the holes was suppressed by the valence band offset between Si and SiGe. As a result. the diffusion mechanisms was dominate factor reducing the retention time.”
(171th line – 177th line, page 9)

Round 2

Reviewer 1 Report

A number of results at different doping levels was added, so the choice seems more reasonable now. However it would be useful to add in lines 191-192 some references to the parameters of the memory cells in terms of a trade-off between sensing margin and retention time.
Additionally, the english in line 73 and in y-axis title in Fig.10 should be checked.

Author Response

Reviewer(s)' Comments to Author:
Comments to the Author
[1] A number of results at different doping levels was added, so the choice seems more reasonable now. However it would be useful to add in lines 191-192 some references to the parameters of the memory cells in terms of a trade-off between sensing margin and retention time.

Answer)
Thank you for your comments. We added references to the parameters of the memory cells in terms of a trade-off between sensing margin and retention time.

 “The variation of Ge content in the proposed 1T-DRAM induced the trade-off between sensing margin and retention time. The 1T-DRAMs obtained the trade-off between sensing margin and retention time due to geometrical parameters, such as channel doping [27] and body thickness [28], which affected simultaneously the stored hole density and recombination rate. Therefore, the memory characteristics of the proposed 1T-DRAMs can be improved by optimizing the geometrical parameters.”
(192th line – 197th line, page 9)
[27] Ansari, M. H. R; Singh, J. Capacitorless 2T-DRAM for High Retention Time and Sense Margin. IEEE Trans. Electron Devices. 2020, 67, pp. 902-906.
[28] Yoon, Y. J.; Cho, M. S.; Kim, B. G.; Seo, J. H.; Kang, I. M. Capacitorless One-Transistor Dynamic Random-Access Memory Based on Double-Gate Metal-Oxide-Semiconductor Field-Effect Transistor with Si/SiGe Heterojunction and Underlap Structure for Improvement of Sensing Margin and Retention Time. J. Nanosci. Nanotechnol. 2019, 19, pp. 6023-6030.
(page 11)

[2] Additionally, the english in line 73 and in y-axis title in Fig.10 should be checked.

Answer)
Thank you for kind comments. We revised the sentence and the y-axis title in Fig. 10.
 “Sentaurus TCAD is capable of simulating numerically electrical and thermal characteristics of devices.”
(73th line – 74th line, page 3)
